# Evaluation of Features Generated by a High-End Low-Cost Electrical Smart Meter

Christina Koutroumpina [1,*], Spyros Sioutas [1], Stelios Koutroubinas [2] and Kostas Tsichlas [1]

[1] Depterment of Computer Engineering & Informatics, University of Patras, 265 04 Patras, Greece; sioutas@ceid.upatras.gr (S.S.); ktsichlas@ceid.upatras.gr (K.T.)
[2] Meazon S.A., Ippodamou Str., 26442, 265 04 Patras, Greece; s.koutroubinas@meazon.com
* Correspondence: koutroumpina@ceid.upatras.gr

**Abstract:** The problem of energy disaggregation is the separation of an aggregate energy signal into the consumption of individual appliances in a household. This is useful, since the goal of energy efficiency at the household level can be achieved through energy-saving policies towards changing the behavior of the consumers. This requires as a prerequisite to be able to measure the energy consumption at the appliance level. The purpose of this study is to present some initial results towards this goal by making heavy use of the characteristics of a particular din-rail meter, which is provided by Meazon S.A. Our thinking is that meter-specific energy disaggregation solutions may yield better results than general-purpose methods, especially for sophisticated meters. This meter has a 50 Hz sampling rate over 3 different lines and provides a rather rich set of measurements with respect to the extracted features. In this paper we aim at evaluating the set of features generated by the smart meter. To this end, we use well-known supervised machine learning models and test their effectiveness on certain appliances when selecting specific subsets of features. Three algorithms are used for this purpose: the Decision Tree Classifier, the Random Forest Classifier, and the Multilayer Perceptron Classifier. Our experimental study shows that by using a specific set of features one can enhance the classification performance of these algorithms.

**Keywords:** energy disaggregation; supervised machine learning; classification





## 1. Introduction

Easy to use and efficient energy disaggregation tools may be the holy grail of energy efficiency [1]. The commercial impact of energy disaggregation at the level of the home customers is the increased utility customer engagement and the reduced energy usage. The goal at this level is to itemize the consumer's energy bill [2], analyze the energy usage and cost per household appliance, make personalized and prioritized energy savings recommendations or even go as far as identify faulty appliances (e.g., frosting cycle of a fridge with a damaged seal is more frequent than a normal one [3]) . All these should be viable through a single sensor per household that monitors the total energy consumption and other related quantities [4]. The requirement in energy disaggregation is to identify the usage of an appliance merely by its signature on the aggregate energy waveform of the household.

The benefits to the consumer from using this technology have already been established [1,3,4]. The social impact of energy disaggregation is reflected on the permanent energy behavior shift of the users, in the reduced communication with the energy provider, in raising the awareness of the users and in providing the opportunity to low-income families to waste less energy. NILM-related products have already started to make their appearance [5], although the market is still at its infancy. Therefore, taking also into account that contemporary appliances can be remotely functioned, an increasing number of people are interested in learning their consumption profile. Moreover, an increasing need is arising for the user to turn from a passive consumer into an active consumer of electricity, who

will be fully aware of what he/she consumes. With respect to the consumer, his/her energy behavior is affected by raising inefficiencies that fall under the following 3 categories:

- **Appliance-Related**: bad performance of an appliance, always-on appliances that are forgotten, low efficiency ratings of appliances, unreasonable large intervals to fall into stand-by-mode;
- **Dwelling-Related**: identification of poor insulation and its effects, geographic position of the dwelling;
- **Energy Use-Related**: how often and when major appliances are run.

To fulfill this, it is necessary to fully record the behavior of all devices used by the user. However, this is not an easy task. One way would be to turn all devices into smart devices, which is very expensive and not environment-friendly and thus quite unlikely to be adopted at least for the near future. Another approach would be to place a smart meter on each device. As smart meters, we consider those which record the energy consumption of an electronic device at regular intervals and allow communication between the utility and the consumer. However, placing a smart meter behind every device is not an efficient way to solve the problem, as it is a difficult and complex process. A more efficient way, as already implied in the previous discussion, is to install a meter on the main panel of the user's home to record the total energy consumption. This is a Non-Intrusive method of Load Monitoring (NILM). In NILM, we attempt to measure total energy rather than using individual meters to measure the energy of each device.

NILM can be roughly decomposed into the following steps (see [3,6,7]):

1. **Load Monitoring—Data Acquisition:** Pre-processing of the raw data and computation of the necessary quantities;
2. **Event Detection:** The steady-state or transient changes that correspond to the operation of appliances are identified. Usually, event-based approaches are employed or state-based approaches [7]. The state-based approach uses each sample from the aggregated signal to detect events. They are mostly based on HMMs and state transitions are obtained automatically from the hidden state estimation. The event-based methods can be further categorized into three classes [8]: expert-heuristic based (state changes are detected by predefined empirical rules), probabilistic model based (e.g.,state changes are detected by probabilistic models—HMMs) and matched-filter based (state changes are detected by signal processing methods);
3. **Feature Extraction:** A set of features are used from the sample to characterize the event. These features constitute the signature of the event. Apart from being hard-wired they can also be extracted automatically by machine learning methods [8];
4. **Classification/Inference:** These signatures are presented to a classification or pattern matching algorithm to identify and assign an appropriate label from the library that will determine the type of the appliance and its state.

The problem of energy disaggregation in NILM (There is a small confusion in the literature as for the terms NILM and energy disaggregation, since most of the time the term NILM contains the notion of energy disaggregation, although the first refers to the method of extracting the signals and the second to the process of breaking down the aggregate signal to the signal of each appliance within the household. We adopt a clear distinction between these two notions) is quite hard since it is an undetermined problem. It becomes even harder since various aspects [6] affect its efficiency and effectiveness: (1) different electrical features, (2) various sampling rates, (3) multiple, simultaneous load events, (4) different appliance types, (5) noisy power signals, (6) dynamic and changing usage, (7) computational cost and complexity, and (8) different accuracy measures. This leads to many problems when actually deploying such a system and the practical limitations of this approach have been identified [9].

## 1.1. Our Contributions

In this paper, we focus on the problem of evaluating different electrical features generated by the smart meter. To this end, we use a specific low-cost and high-performance smart meter produced by Meazon S.A. (Meazon S.A. has a major role in the market in the field of energy efficiency [10]. The company's main mission is the design and manufacturing of low-cost and high-performance energy meters of small size). This smart meter was installed on the main panel of a house. The relatively high sampling rate (compared to its low cost) as well as the richness of the generated features by this smart meter enable us to look at how incorporating various features in energy disaggregation methods affect their performance. To this end, we use a simple setting for load identification and use three different basic machine learning methods for energy disaggregation from the aggregate signal. Our goal is to understand how the accuracy of the predictions of the methods are affected by incorporating more features, thus raising the dimensionality of the problem and its respective time complexity. The rather medium sampling rate of the smart meter (50 Hz) allows us to capture better the transient behavior of the appliances when starting their operation and thus the multidimensional signal supports better results at disaggregating appliances without requiring expensive equipment and complicated circuitry. Although our results focus on the specific smart meter, their generalization to other smart meters of similar characteristics is immediate. Compared to previous work, our contribution lies at the fact that we look at power-related features that are generated from the smart meter without any processing from the client side. In addition, we look at a rather unexplored area with respect to sampling rate, which is set at 50 Hz. Our long-term goal is to use these features for energy disaggregation on the smart meter by adopting a streaming approach. This means that we will target energy disaggregation methods that use minimal memory and can process each measurement in minimal time in order to report the active appliances. This paper is a first small step towards this goal by pinpointing the set of features to use.

## 1.2. Related Work

The field of NILM and energy disaggregation is very extensive and spans different scientific areas. To understand various aspects of NILM (from electrical characteristics of appliances to data requirements and algorithms for appliance identification), some good points to start from are [7,11–18]. A small note on the sampling rate of the publicly available datasets is in order. A very recent review [19] provides a description of well known publicly available datasets for energy disaggregation, most of which are either of low frequency ($\leq$1 Hz) or of high frequency ($\geq$10 KHz), with minimal representative datasets in the range of 10 Hz to 500 Hz. This is why there are not many results for NILM in this frequency range. In the following, we discuss only the literature that is straightforwardly related to the results of this paper.

The features are assumed to be chosen beforehand and not extracted automatically by machine learning methods as in [8]. An extensive discussion with respect to different features for energy disaggregation can be found in [5,20]. Multivariable (many features) approaches have been used for various stages of energy disaggreggation (e.g., see [21,22] for event detection) aiming at improving effectiveness at the expense of raising the dimensionality of the problem and thus aggravating its efficiency.

The impact of using more features for energy disaggregation is usually explored within a proposed framework for solving energy disaggregation or related energy problems. For example, in [23], the authors show how different features improve on the results of the identification of the chosen TV channel based only on the consumption of the TV. In [24], the authors extend matrix factorization to many features (active, reactive, apparent, and current) and show that using more features provide better results than the one-feature approach in [25]. Additionally, the book in [18] contains experiments about the evaluation of features when combined with specific approaches for energy disaggregation.

To the best of our knowledge, there are not many results related solely to general feature evaluation for energy disaggregation. In [26] the authors discuss the features that are usually used in NILM up to 2017, and describe an algorithm for selecting the best set of features. In particular, they use Random Forest as a classification method, and apply it on a dataset collected at a 30 KHz sampling rate. They look at power features (e.g., active power, RMS current, etc.) as well as spectral features (e.g., $j$-th current harmonic coefficient) both for steady state as well as for transient. They describe an algorithm that automatically selects the feature set that seems to have the best performance. In [27] they evaluate four different machine learning methods for seven different feature sets, showing that Random Forest gives the best results. The datasets they use were collected at 1 Hz and their results show that among the seven different feature sets they use, the best depend on the type of the appliance (resistive/inductive/non-linear). The feature sets consist of statistical features (e.g., mean power) and electrical features (e.g., load angles) showing that the latter contribute more to the differentiation between distinct appliances. The statistical features were generated over a time window of 10 samples (10 s). Additionally, in [18] a preliminary evaluation of certain features is performed on a 44 KHz dataset, leading to the conclusion that the active power and reactive power have quite high differentiation power among many different appliances at this sampling frequency.

Finally, as previously mentioned, our goal is to use energy disaggregation methods that have minimal memory and CPU requirements since we would like them to run on the smart meter. Some steps have been taken towards this direction. In [28], a supervised HMM (Hidden Markov Model) method that exploits the sparsity of such models is presented, which according to the authors is so efficient that it could run on an embedded processor. Another recent result based on Factorial HMMs is in [29]. They state that the complexity of their approach is linear to the number of events and thus it is amenable for implementation at the edge of the cloud. Similarly, in [30,31] lightweight machine learning techniques are presented that allegedly can be run on small microprocessors. An unsupervised combinatorial algorithm is analyzed in [32]. They first determine the features of appliances and then they use a triangle-rectangle decomposition of the aggregate signal leading to a lightweight and real-time approach.

In the following section we provide basic definitions as well as a description of the capabilities of the smart meter. In Section 3 we discuss our experimental setup and our methodology with respect to the experimental evaluation while in Section 4 we provide our experimental findings with respect to how different features affect the identification of a single appliance from the aggregate signal. Finally, we conclude in Section 5 with some remarks and extensions.

## 2. Preliminaries

Assume that for the time instances $\mathcal{T} = \{1, \ldots, T\}$ we get the set of measurements on the aggregate power consumption $\mathcal{X} = \{x_1, x_2, \ldots, x_t\}$ from the set of active appliances $\mathcal{A} = \{1, \ldots, n\}$. The task of NILM is to infer the power contribution $y_t^i$ of appliance $i \in \mathcal{A}$ at time $t$ such that at any point in time it holds that:

$$x_t = \sum_{i=1}^{n} y_t^i + \sigma(t), \tag{1}$$

where $\sigma(t)$ represents the contribution from unknown appliances or from noise.

To evaluate the various features provided by the smart meter we use some basic machine learning algorithms. In order to evaluate their effectiveness we use standard metrics that we describe briefly in the following. We use accuracy, recall, precision, and the $F_1$-score. Accuracy is defined as the number of correct predictions divided by the total number of predictions.

$$Accuracy = \frac{TP + TN}{TP + FP + FN + TN}, \tag{2}$$

where $TP$ is the number of True Positives, $FP$ is the number of False Positives, $TN$ is the number of True Negatives, and $FN$ is the number of False Negatives. Recall is the number of true samples that are successfully classified over the sum of the number of true positives and the number of false negatives.

$$Recall = \frac{TP}{TP + FN}. \tag{3}$$

Precision is the number of true records that are successfully classified over the sum of true and false positives.

$$Precision = \frac{TP}{TP + FP}. \tag{4}$$

The $F_1$-score is the harmonic mean of recall and precision.

$$F_1 = 2 * \frac{precision * recall}{precision + recall}. \tag{5}$$

### 2.1. Din-Rail Meter and Its Interface

Our experimental setup basically consists of a three-phase electricity meter designed by Meazon S.A., which is installed in the central electricity panel of a house. This meter is capable of remote monitoring and controlling the energy consumption of a household and/or an industrial building [33]. It is a rail-type device with small size (2 DIN), which can be easily installed and implements monitoring, measurement logging and controlling of:

- Up to three separate power lines/loads;
- A three-phased load;
- An entire electrical panel (single-phase, dual-phase, three-phase).

The meter was selected as it offers accurate measurements and a detailed profiling of energy load. It supports a wi-fi connection in order to communicate the data generated 50 times per second. Meazon's technology incorporates all the signal processing on the smart meter required to measure the following variables (features):

- RMS Voltage

$$V_{RSM} = \sqrt{\frac{1}{T_s} \times \int_0^{T_s} V(t)^2 dt}, \tag{6}$$

- RMS Current

$$I_{RMS} = \frac{I_{max}}{\sqrt{2}}, \tag{7}$$

for sinusoidal currents. The meter also computes the RMS values of various fundamental and harmonic components of phase currents, phase voltages, and neutral current as part of the harmonic calculations.

- Active power

$$P = V \times I \cos(\varphi), \tag{8}$$

where $\varphi$ is the phase angle between vectors $V$ and $I$.

- Reactive power

$$Q = V \times I \sin(\varphi), \tag{9}$$

where $\varphi$ is the phase angle between vectors $V$ and $I$.

- Apparent power

$$S = V \times I \tag{10}$$

- Line frequency. Stable on 50 Hz.
- Power factor

$$PF = \cos \varphi \tag{11}$$

- Crest factor

$$CF = \frac{|\text{Peak Current}|}{\text{RMS Current}} \tag{12}$$

- Angle between *V* and *I*

The above equations were taken from the Evaluation Board User Guide [34]. Apart from these variables, it also computes energy-related quantities as well as the RMS of harmonics on the phase and the neutral currents and on the phase voltages, together with active, reactive and apparent powers, and the power factor and harmonic distortion on each harmonic for all phases. Total harmonic distortion (THD) is computed for all phases for current and voltage. Another distinctive characteristic of the particular low-cost smart meter when compared to others is that the interval between successive measurements (the sampling rate) is at 20 ms (50 Hz). In Figure 1, two different devices are measured with 1 s interval, which is usually the smallest in the literature for low-cost smart meters, and 20 ms interval. It is clear that with 1 Hz sampling rate important information related to the transients is lost.

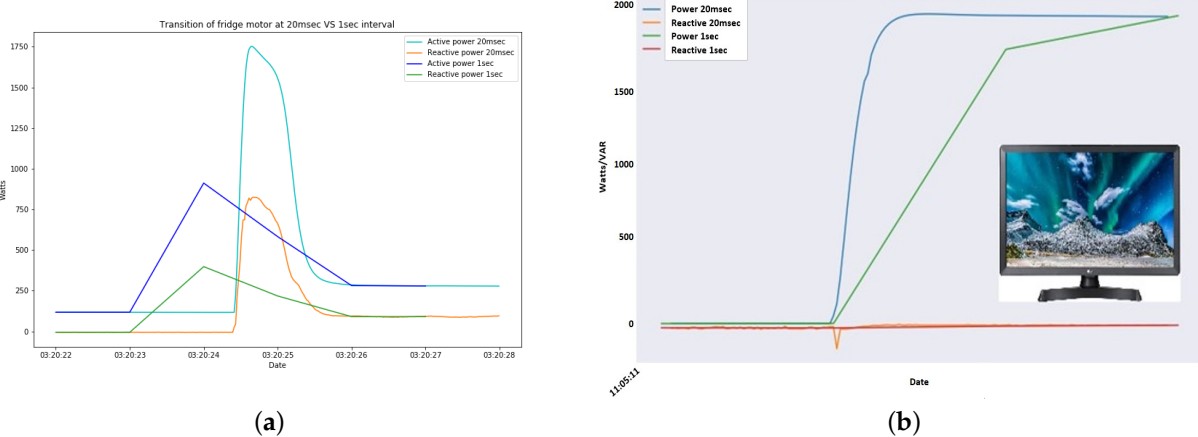

(**a**)　(**b**)

**Figure 1.** Comparing time intervals. (**a**) Measuring fridge's consumption; (**b**) Measuring TV's consumption.

All data generated by the smart meter are stored and processed in a Cassandra database on Azure. This is achieved by an application developed by Meazon that can run on any computer system although for the project needs, this application is integrated to a small single board computer called Janus. Janus has the necessary functionality to run this application and send data on Azure. The application uploads the measurements in the cloud and allows users to analyze the data in a dashboard. In Figure 2, the data flow from the smart meter to the user's screen is depicted.

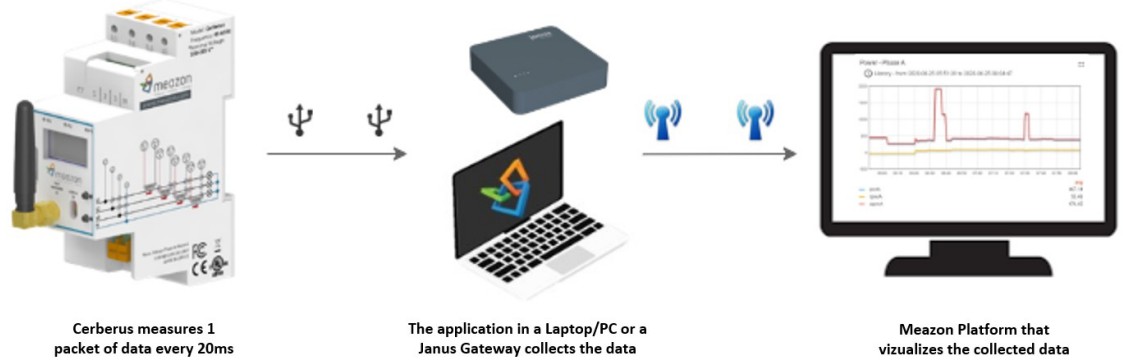

**Figure 2.** Data flow from the smart meter to the user's screen.

### 2.2. The Online Platform

The online platform allows the user to visualize the collected data. After successfully logging in to this platform, the user has access to the folders HOME, DEVICES, ASSETS, ENTITY VIEWS, and DASHBOARDS. The HOME folder is shown in Figure 3 and constitutes the central page of the online platform from which access is provided to other folders.

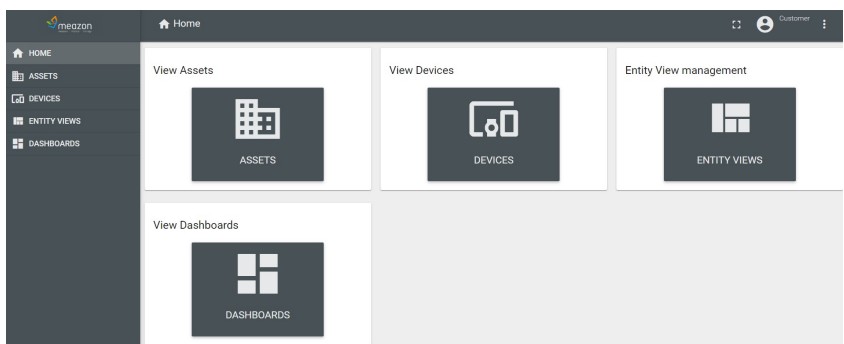

**Figure 3.** Home view.

In the DEVICES folder, the user has access to all Meazon's devices that have connected to the online platform. These devices include the smart meter(s) in the central electrical panel(s) as well as smart plugs connected to electrical sockets that measure particular devices connected to these sockets (e.g., a fridge). In Figure 4, the serial numbers of two different active meters are depicted.

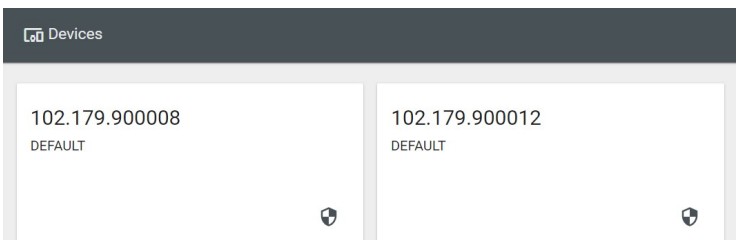

**Figure 4.** Devices view.

The DASHBOARD folder contains all the visualizations of the variables generated by the smart meter in a simple user interface easily accessible from a computer or a mobile device. The data may be visualized in real time (see Figure 5) or they may be retrieved for an arbitrary time period (see Figure 6).

The dashboard also allows the user to construct small but valuable ground truth data that corresponds to events related to a period of time during which a particular appliance is ON and working. More precisely, the user is allowed to enter details related to the start and end time, appliance information, electrical information, modes of operations and other various comments as shown in Figure 7. The aforementioned events are stored in the database and can be retrieved from another widget. There are various filters that can be applied in a query regarding the device's name, starting time, event's duration, serial number, device type, manufacturer and more (Figure 8). Finally, assets are abstract IoT entities that can be related to other devices and assets while entity views are used to provide access to certain aspects of a device or an asset to a customer.

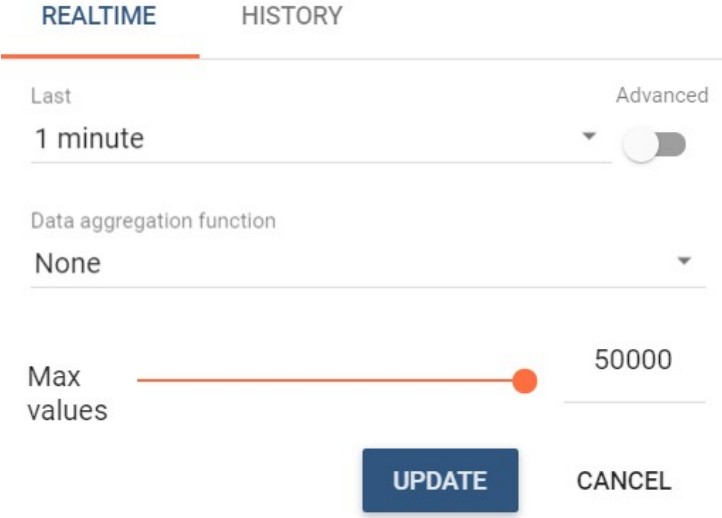

**Figure 5.** Real-time data.

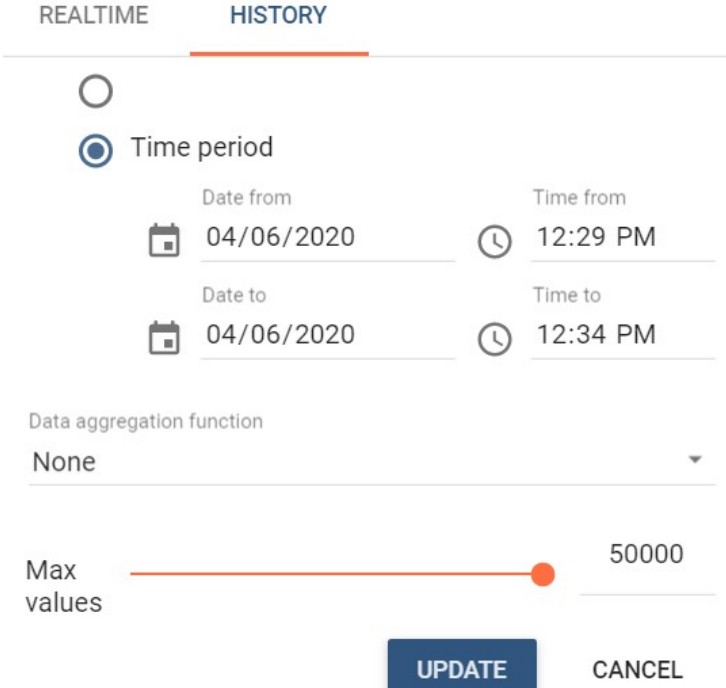

**Figure 6.** Historical data.

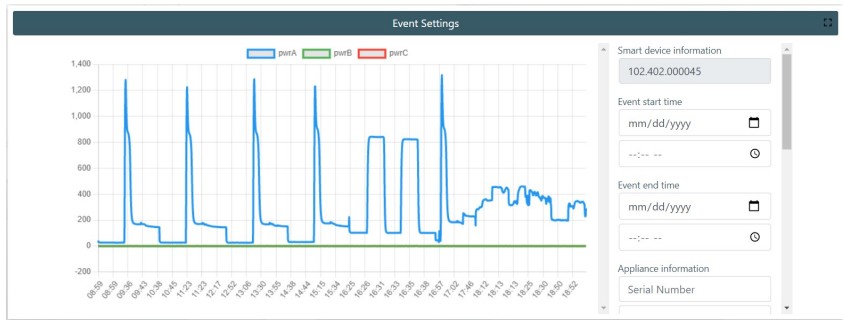

**Figure 7.** Setting event parameters.

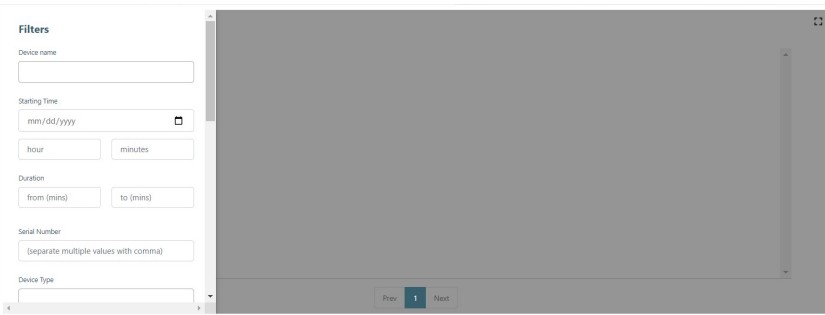

**Figure 8.** Filters.

## 3. Experimental Methodology and Setup

Our approach is to use appliance-specific models that classify the input into two classes. The first is denoted by "1" and indicates that in this measurement the particular device is operating. The second class is denoted by "0" and indicates that the device is off. Each device has its own classifier. Each classifier is trained with a different file, prepared for the specific device. The features we use to train the classifiers are generated by the smart meter, and thus no feature extraction is required. In particular, the features that are available during training are "active power", "apparent power", "reactive power", "current", "angle between $V\&I$", "crest factor" and the binary variable "output".

The classifiers that were examined in order to solve the binary classification problem are the Decision Tree Classifier, the Random Forest Classifier, and the Multilayer Perceptron Classifier. For the Random Forest classifier, we set the number of trees to 10 but we leave the maximum tree depth undefined. For the Multilayer Perceptron Classifier, the input layer was set to size 6 as we are considering at most 6 features, the two intermediate layers were set to 5 and 4 and the output layer has size 2 as there are 2 available output classes. For stacking the input data in matrices, the block size was set to 128. The seed for weights initialization was "System.currentTimeMillis", which is a method that returns the current time in milliseconds. Finally, the convergence tolerance of iterations is set to 200. In order to test our classifiers, we used 3-fold cross validation in order to balance performance and variance in the results.

### 3.1. Data Retrieval

The retrieval of the data from the platform is performed through HTTP Requests. The features generated by the smart meter that are used for training the models are the following:

- pwr: Active power;
- apwr: Apparent power;
- rpwr: Reactive power;
- cur: Current;
- angle: Angle between $V\&I$;
- scre: Crest factor.

These features are selected because of their importance in identifying appliances. Other features generated by the smart meter, such as energy, may be useful for different kinds of problems, such as regression. Additionally, the line voltage and the line frequency remain almost fixed and thus do not contribute much to the differentiation among appliances.

To export the data, a Jupyter Notebook file was created. First, the POST HTTP method is invoked, which sends data to a server to create a resource. To request the data from the specified resource, we use the HTTP method GET parameterized by the address of the website, the identity of the smart meter, the features we are interested in as well as the time interval that interests us. The online platform is tuned so that the user can export and store up to 50.000 measurements at a time. This number corresponds to approximately 15 min of measurements and therefore, if the user chooses to retrieve data for a bigger

amount of time, this interval is divided into subsections of 15 min and the retrieval process is repeated for each of these subranges. The retrieved data are saved in a dataframe, which is then converted to a CSV file. Five appliances were selected for experimentation. These devices are quite common in households and correspond to a considerable part of the total consumption of a typical house. These are a vacuum machine, an electric oven, an air-condition, an electric stove, and an ironing press. To measure the appliances during their function, we used only the din-rail meter and identified a time period when no other device changed its status of operation in the house. We adopted this approach since we did not want to use additional smart plugs that would provide us with a clean and simple signal of each appliance. Each appliance was operated for 15 min and then it was turned off. This means that the data used by the models, were obtained from different time periods that were not continuous. Figures 9–18 depict the consumption behavior of each appliance.

In Figure 9, the active/reactive and apparent power of the vacuum cleaner is depicted during the start of the operation focusing at the transient behavior until the steady state is reached. In Figure 10, the total active/reactive and apparent power is depicted during the 15 min interval that the vacuum cleaner is functioning. In Figures 11–18 the same quantities are depicted for the the other devices as well.

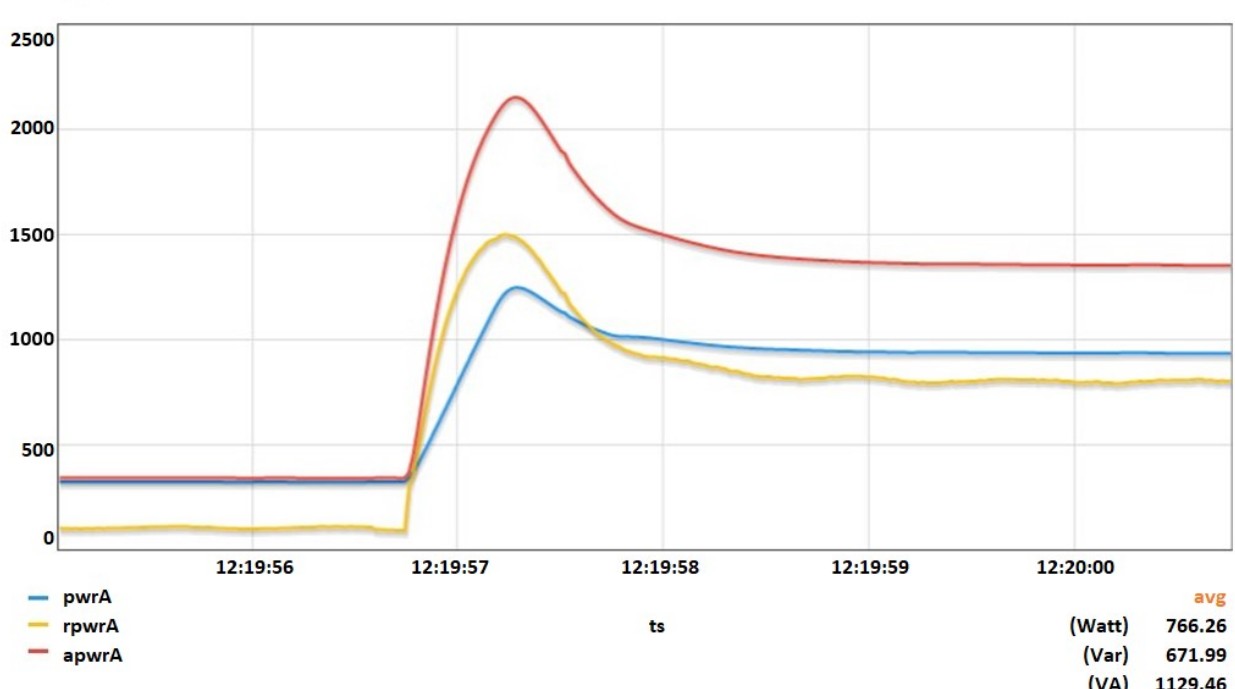

**Figure 9.** A snapshot from vacuum's opening.

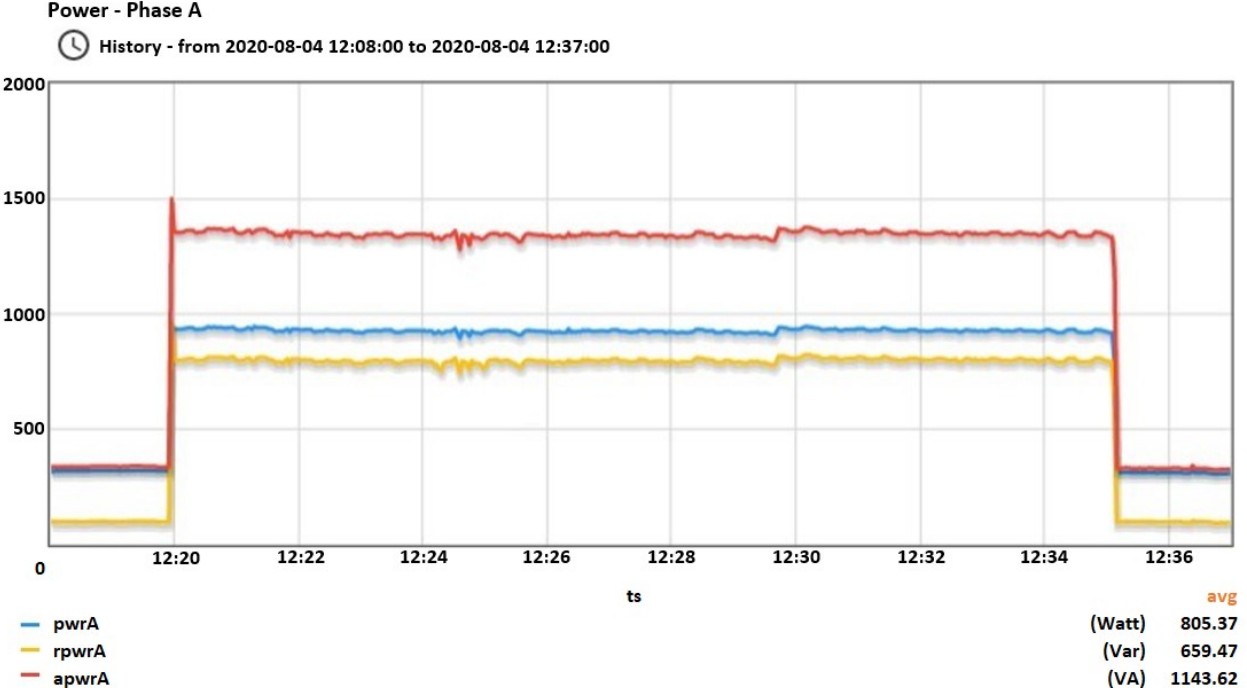

**Figure 10.** Vacuum's total consumption from the beginning of its function until the end. Vacuum operated in a time period where no other device changed the state.

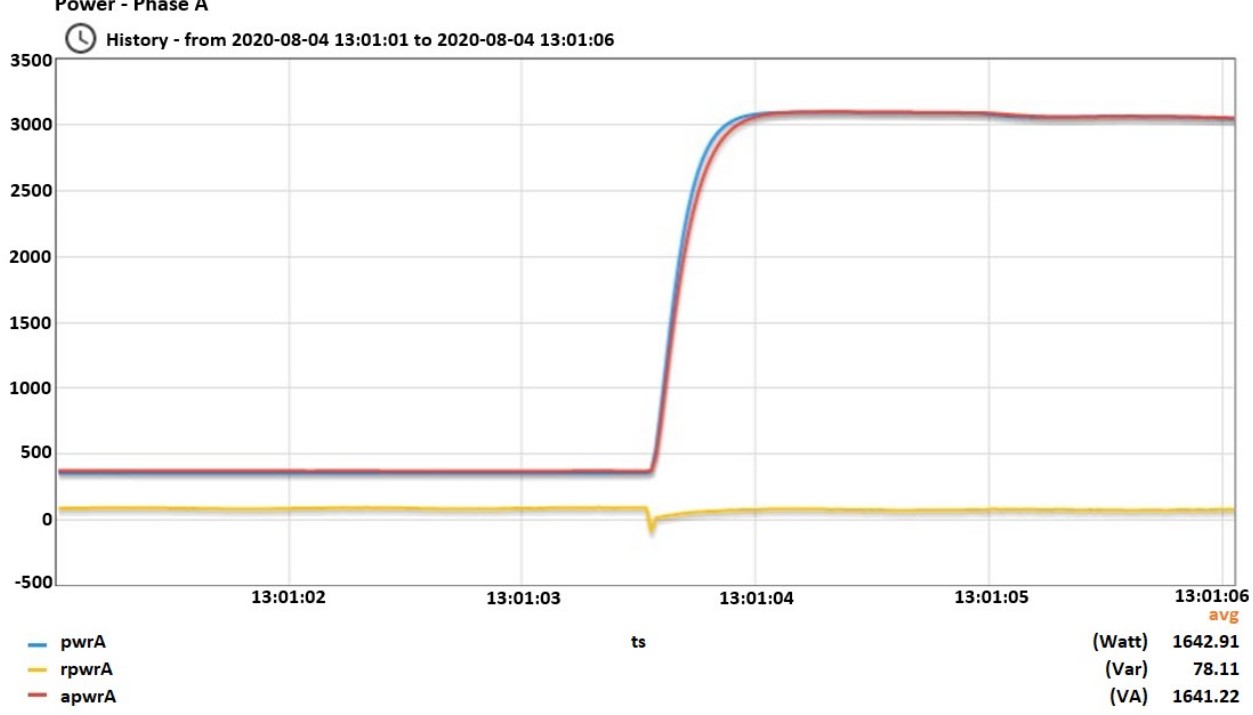

**Figure 11.** Oven's opening.

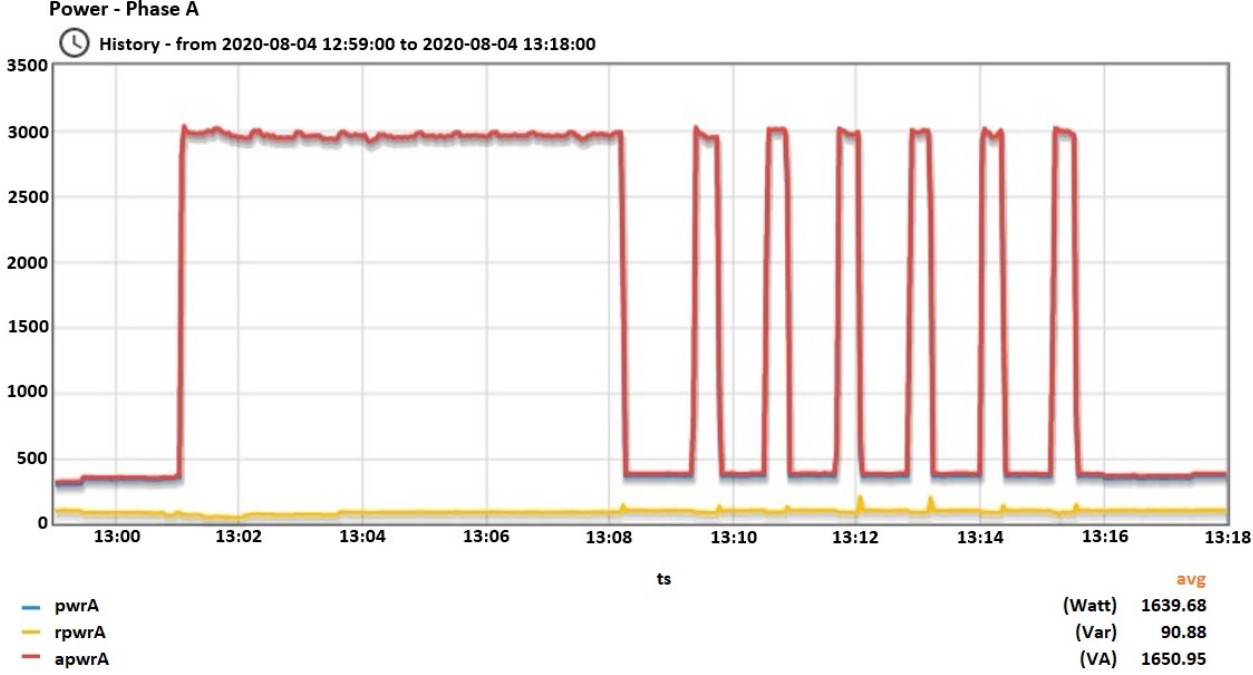

**Figure 12.** Oven's behavior. The oven operated in a time period where no other device changed the state.

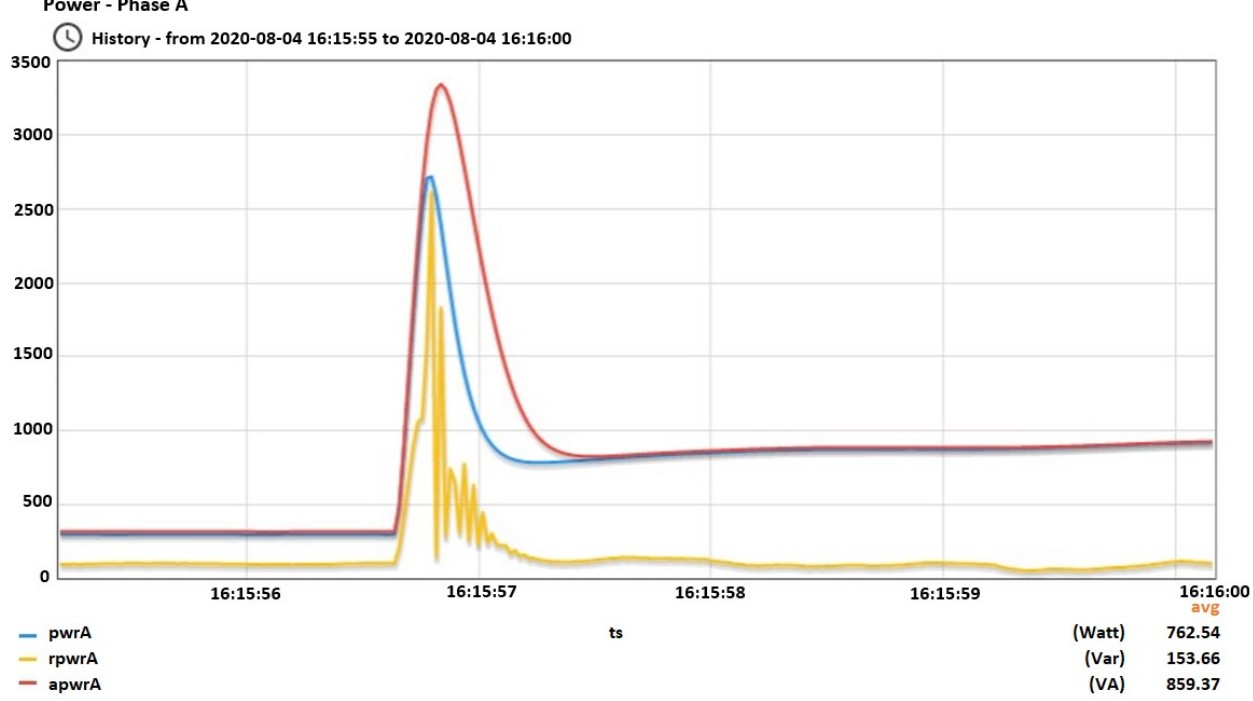

**Figure 13.** Air-conditioner's opening.

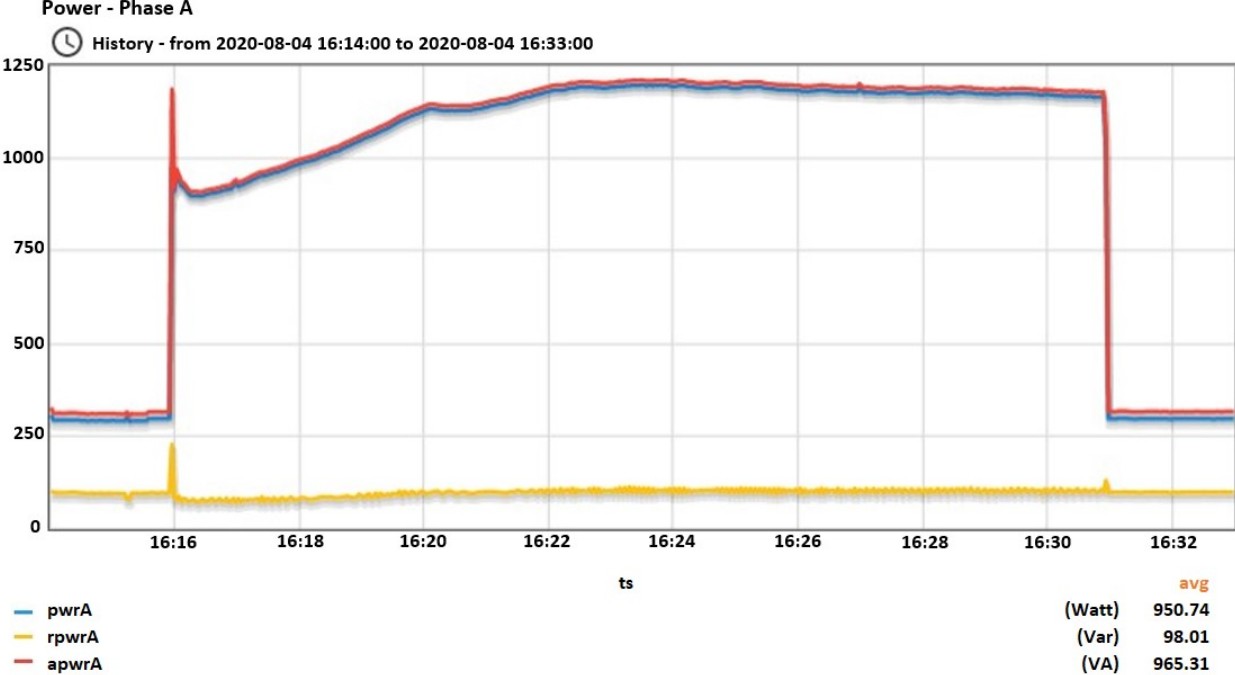

**Figure 14.** Air-conditioner's behavior. The air-conditioner operated in a time period where no other device changed the state.

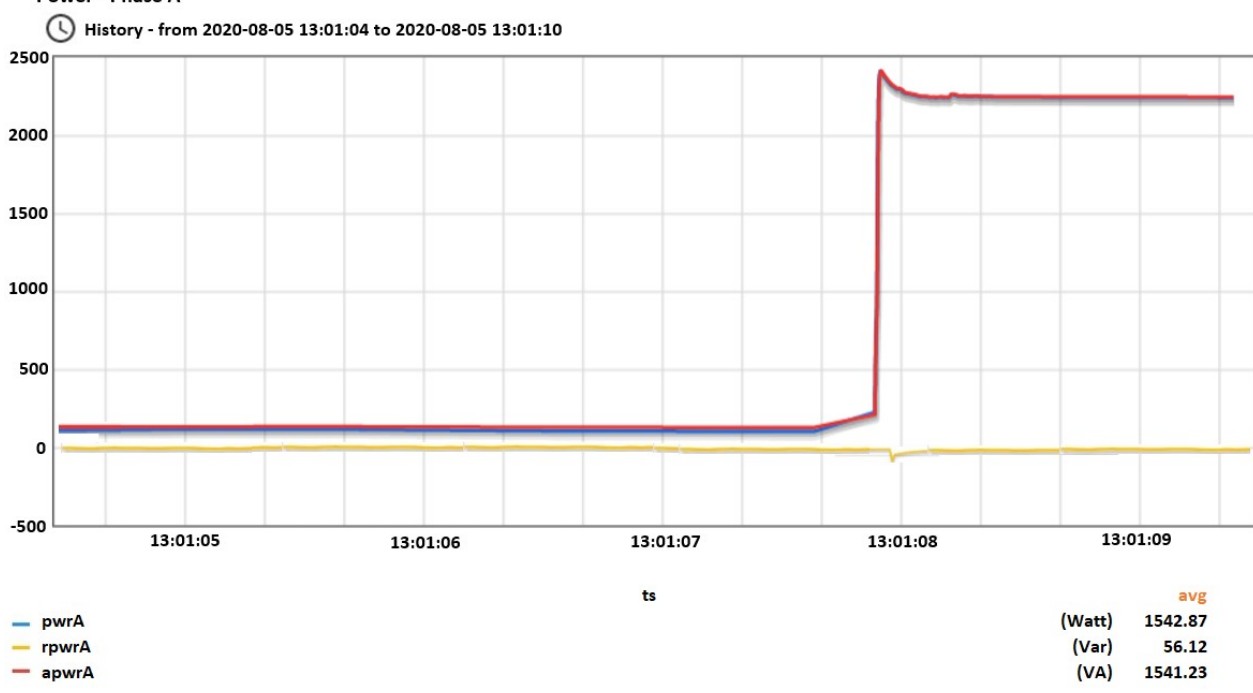

**Figure 15.** Stove's opening.

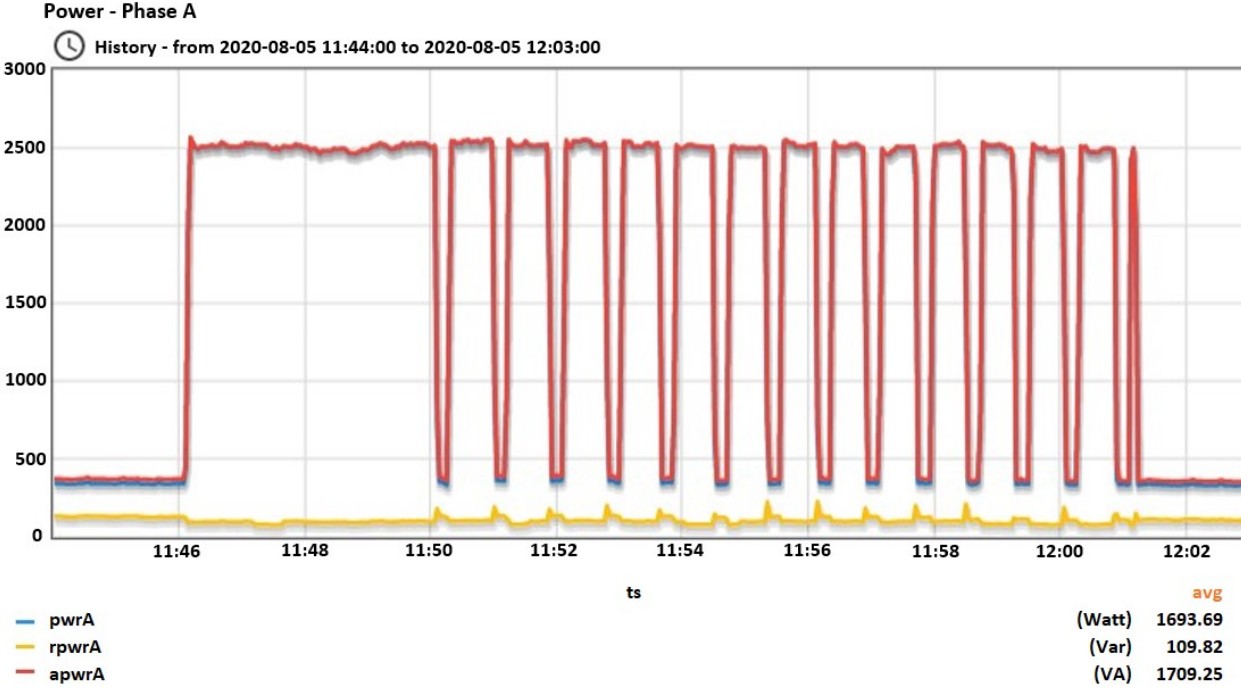

**Figure 16.** Stove's behavior. The stove operated in a time period where no other device changed the state.

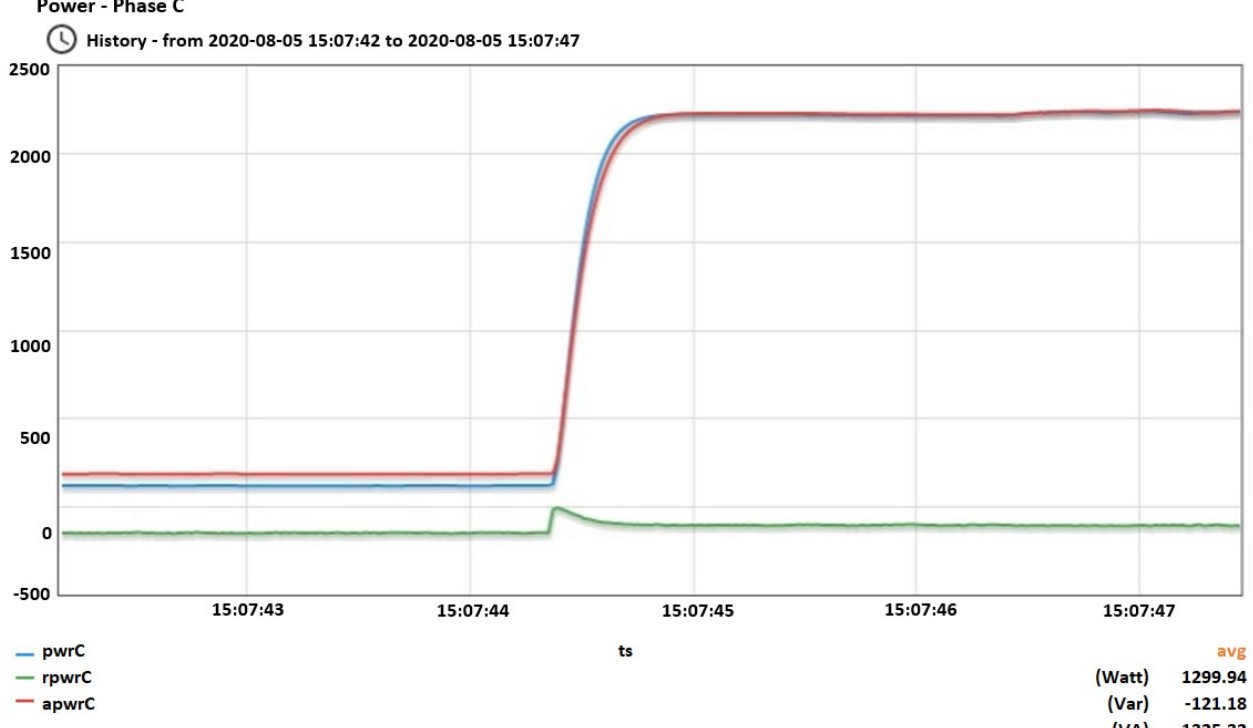

**Figure 17.** Press' opening.

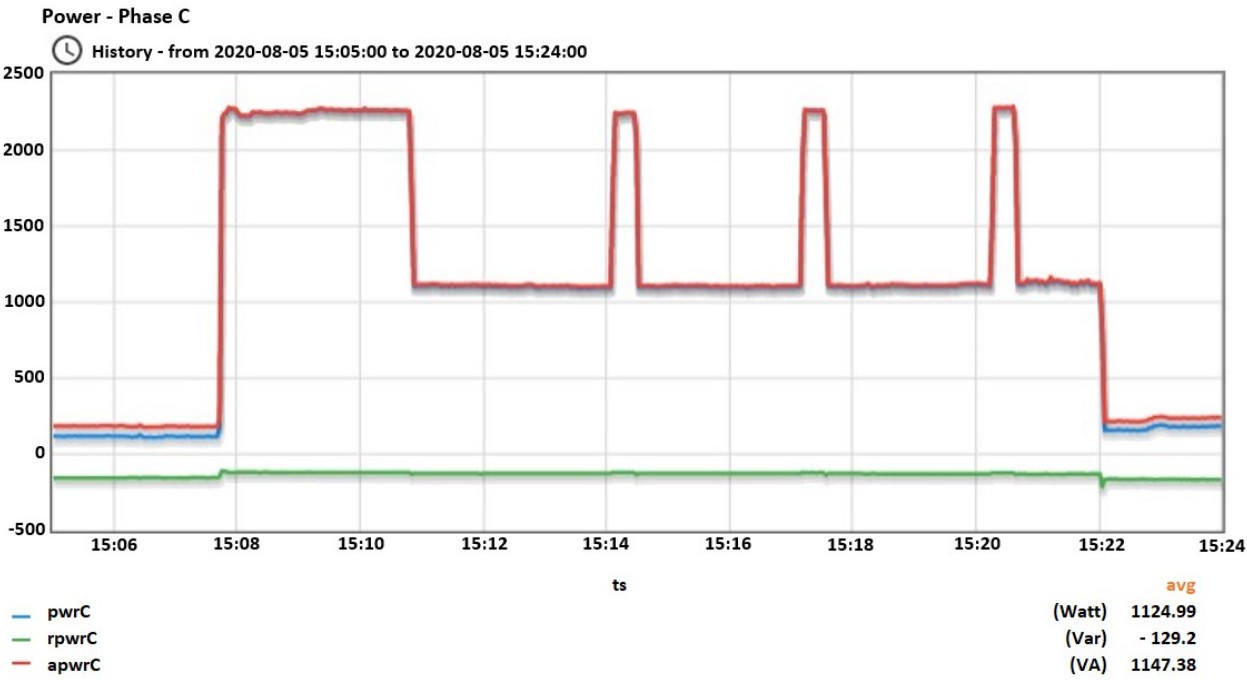

**Figure 18.** Press' behavior. The ironing press operated in a time period where no other device changed the state.

### 3.2. Data Preprocessing

After the collection of the measurements is completed, we preprocess them in order to make them suitable for the training of the machine learning models. An extra column is created, the "output" column. This column represents the class in which each measurement belongs, with respect to the function of the corresponding appliance. Thus, there are two classes available, the "0" and "1". When the measurement is assigned in class "0", it means that this device is inactive at the specified time. On the other hand, when the measurement is assigned in class "1", it means that the device is active at the specified time. Each measurement is placed in the class that it belongs, by observing the values of the other attributes.

### 4. Experimental Results

The experiments were performed on a 64-bit operating system with 8GB RAM, 4 cores and an Intel Core i7-8565U CPU processor. The time needed for the execution of the Decision Tree classifier was 2 min on average. This was also the case for the Random forest classifier. On the contrary, the Multilayer Perceptron classifier needed 5 min on average.

Our experimental design is based on trying out subsets of the set of acquired features in order to compare them with the full set with respect to their classification effectiveness. In particular, for the Decision Tree and Random Forest classifiers we used only active power compared to the full feature set while for the Multilayer Perceptron Classifier, we used both active and reactive power since otherwise the results were quite poor. In all cases, the classification effectiveness of the classifiers was worse that the case where all the features are used. The results can be found in Tables 1–7. In our experiments, we notice that most of the measurements that were classified wrongly were those in the transient phase, that is during the first milliseconds of the operation of the appliance. The measurements during the transient phase are not easy to classify. However, by adding more features in the training phase, we manage to wrongly classify less measurements during the transient phase. This means that the additional information provided in the form of additional features helps in correctly identifying the operating appliance. In the following, we discuss the results for each appliance.

For the vacuum cleaner (see Table 1), both for the Decision Tree classifier and the Random Forest classifier, the results are improved considerably when using the whole set

of features instead of the active power alone. However, the improvement in the Multilayer Perceptron classifier was only marginal. This is due to two reasons: on the one hand, this classifier compares active-reactive to the whole set of features and on the other hand the vacuum cleaner has one mode of operation and thus it is easy for this classifier to identify the measurements. The improvement of the classification results were mainly located at the transient phase.

**Table 1.** Results for the vacuum cleaner. With "A" we refer to the experiments that used only active power. With "AR" we refer to experiments with active and reactive power and with "A+" we refer to experiments with all the selected attributes.

| Vacuum | Decision Tree Classifier | | Random Forest Classifier | | Multilayer Perceptron Classifier | |
|---|---|---|---|---|---|---|
| | **A** | **A+** | **A** | **A+** | **AR** | **A+** |
| Accuracy | 0.974 | 0.9999 | 0.9738 | 0.9999 | 0.9994 | 0.9998 |
| Recall | 0.9023 | 0.9997 | 0.9292 | 0.9999 | 0.9991 | 0.9993 |
| Precision | 0.9312 | 0.9998 | 0.9046 | 0.9995 | 0.9979 | 1 |
| F-Measure | 0.9165 | 0.9998 | 0.9167 | 0.9997 | 0.9986 | 0.9996 |

Regarding the electrical oven, its behavior is similar to the behavior of the stove and as a result we discuss them together (see Tables 2 and 3). It is more difficult to correctly classify measurements from those devices due to the many changes in the power consumption. Furthermore, the Multilayer Perceptron classifier has surprisingly less accurate predictions than the rest of the classifiers. Yet, we can observe that with the extra features, the results are noticeably better, for all the classifiers. Once more, we observed that the improvement in the classification results when using all features was mainly located in the transient phase.

**Table 2.** Results for the oven. With "A" we refer to the experiments that used only active power. With "AR" we refer to experiments with active and reactive power and with "A+" we refer to experiments with all the selected attributes.

| Oven | Decision Tree Classifier | | Random Forest Classifier | | Multilayer Perceptron Classifier | |
|---|---|---|---|---|---|---|
| | **A** | **A+** | **A** | **A+** | **AR** | **A+** |
| Accuracy | 0.9804 | 0.9902 | 0.9713 | 0.9907 | 0.8791 | 0.8891 |
| Recall | 0.9447 | 0.9865 | 0.9043 | 0.9565 | 0.3206 | 0.3497 |
| Precision | 0.9267 | 0.9516 | 0.9083 | 0.9843 | 0.7247 | 0.8203 |
| F-Measure | 0.9356 | 0.9687 | 0.9063 | 0.9702 | 0.4446 | 0.4863 |

**Table 3.** Results for the stove. With "A" we refer to the experiments that used only active power. With "AR" we refer to experiments with active and reactive power and with "A+" we refer to experiments with all the selected attributes.

| Stove | Decision Tree Classifier | | Random Forest Classifier | | Multilayer Perceptron Classifier | |
|---|---|---|---|---|---|---|
| | **A** | **A+** | **A** | **A+** | **AR** | **A+** |
| Accuracy | 0.8624 | 0.9871 | 0.8809 | 0.9839 | 0.6639 | 0.6624 |
| Recall | 0.8609 | 0.9779 | 0.8984 | 0.9769 | 0.7659 | 0.987 |
| Precision | 0.7869 | 0.9871 | 0.7645 | 0.9795 | 0.533 | 0.5235 |
| F-Measure | 0.8222 | 0.9825 | 0.8261 | 0.9782 | 0.6286 | 0.6841 |

As for the ironing press and the air-conditioner, the differences between implementing the training with only active power and implementing the training with the whole set of features are shown in Tables 4 and 5 and are obviously better for the latter set of features. For the ironing press we delve deeper by looking at various subsets of features. The results can be found in Tables 6 and 7. In general, when adding more features to the training phase, the classifiers return less false negatives and false positives. For example, when using active power, reactive power and crest factor, the metrics return better values, although they can be further enhanced. In the case of considering four features, such as active power,

reactive power, RMS current, and crest factor, the results are closer to the optimal. The combination of features that show the best results and can match the results of the whole set of features is active power, reactive power, angle between *V&I*, and crest factor. It seems that the ironing press' behavior could be better classified, using only these four parameters. In general, the same behavior was observed in the other appliances as well.

**Table 4.** Results for the air-conditioner. With "A" we refer to the experiments that used only active power. With "AR" we refer to experiments with active and reactive power and with "A+" we refer to experiments with all the selected attributes.

| A/C | Decision Tree Classifier | | Random Forest Classifier | | Multilayer Perceptron Classifier | |
|---|---|---|---|---|---|---|
| | **A** | **A+** | **A** | **A+** | **AR** | **A+** |
| Accuracy | 0.9293 | 0.9997 | 0.9417 | 0.9996 | 0.9339 | 0.9987 |
| Recall | 0.662 | 0.9995 | 0.8994 | 0.999 | 0.6641 | 0.9976 |
| Precision | 0.8574 | 0.9988 | 0.71 | 0.9989 | 0.8907 | 0.9938 |
| F-Measure | 0.7471 | 0.9992 | 0.7936 | 0.999 | 0.7609 | 0.9968 |

**Table 5.** Results for the ironing press. With "A" we refer to the experiments that used only active power. With "AR" we refer to experiments with active and reactive power and with "A+" we refer to experiments with all the selected attributes.

| Press | Decision Tree Classifier | | | Random Forest Classifier | | | Multilayer Perceptron Classifier | |
|---|---|---|---|---|---|---|---|---|
| | **A** | **AR** | **A+** | **A** | **AR** | **A+** | **AR** | **A+** |
| Accuracy | 0.9573 | 0.9965 | 0.9999 | 0.959 | 0.9982 | 0.9999 | 0.9998 | 0.9998 |
| Recall | 0.8854 | 0.9984 | 0.9995 | 0.7908 | 0.9899 | 0.9999 | 0.9992 | 0.9993 |
| Precision | 0.8384 | 0.9787 | 0.9999 | 0.9891 | 0.9982 | 0.9997 | 0.9997 | 0.9992 |
| F-Measure | 0.8613 | 0.9884 | 0.9997 | 0.8789 | 0.9941 | 0.9998 | 0.9994 | 0.9993 |

**Table 6.** Results from the press' classifiers with various subsets of features used for training. With "A, R, Ap, Cr" we denote the features Active power, Reactive power, Apparent power, and Crest factor respectively.

| Press | Decision Tree Classifier | | Random Forest Classifier | | Multilayer Perceptron Classifier | |
|---|---|---|---|---|---|---|
| | **A, R, Ap** | **A, R, Cr** | **A, R, Ap** | **A, R, Cr** | **A, R, Ap** | **A, R, Cr** |
| Accuracy | 0.9996 | 0.9994 | 0.9996 | 0.9996 | 0.9339 | 0.9511 |
| Recall | 0.9976 | 0.9977 | 0.999 | 0.999 | 0.9981 | 1 |
| Precision | 0.9998 | 0.9985 | 0.9984 | 0.9989 | 0.9999 | 0.7546 |
| F-Measure | 0.9988 | 0.9981 | 0.9987 | 0.999 | 0.9996 | 0.8601 |

**Table 7.** More results from the press' classifiers with various subsets of features used for training. With "A, R, Cu, Cr, An" we denote the features active power, reactive power, RMS current, Crest factor, and Angle between *V* and *I*.

| Press | Decision Tree Classifier | | Random Forest Classifier | | Multilayer Perceptron Classifier | |
|---|---|---|---|---|---|---|
| | **A, R, Ap** | **A, R, Cr** | **A, R, Ap** | **A, R, Cr** | **A, R, Ap** | **A, R, Cr** |
| Accuracy | 0.9996 | 0.9994 | 0.9996 | 0.9996 | 0.9339 | 0.9511 |
| Recall | 0.9976 | 0.9977 | 0.999 | 0.999 | 0.9981 | 1 |
| Precision | 0.9998 | 0.9985 | 0.9984 | 0.9989 | 0.9999 | 0.7546 |
| F-Measure | 0.9988 | 0.9981 | 0.9987 | 0.999 | 0.9996 | 0.8601 |

Finally, we discuss in detail the distribution of faulty classification measurements in the case of the ironing press. In the case where only active power is considered, the Random Forest classifier had to classify 17.231 records in class "1" (ground truth) and returned 17.104 true positives, 127 false negatives and 4476 false positives. A total of 37% of false negatives are located in the transient phases. Furthermore, 97% of false positives are found during the operation of the electric stove. This is because those two devices have

very similar behavior. Finally, when using all the selected features, the Random Forest classifier had 17.256 true positives, 4 false negatives and 1 false positive. 3 out of 4 false negatives are located in the transient phases of the operation of the ironing press, while the only false positive was again found during the operation of the stove.

In summary, the experimental results show clearly that by using more features the effectiveness of the chosen methods is improved. It is not imperative to use the whole set of six features, but as previously mentioned, using a subset of four features provides results that almost match the results of the full set. As a byproduct, it is observed that the Random Forest classifier (in accordance to [27]) shows the best results for classification followed by the Decision Tree classifier with somewhat worse results. Finally, the Multilayer Perceptron classifier has the worst performance. For this classifier, the results were quite bad for the electric stove and the electric oven, which have similar power consumption. On the contrary, for devices that show unique behavior, such as the vacuum cleaner, the air-conditioner, and the ironing press, all classifiers returned very good results.

## 5. Conclusions and Future Work

The developed system categorizes measurements that arise from the total energy consumption of a house into two classes per appliance (either ON or OFF). To achieve this, only one smart meter is used and installed on the main panel without using any additional plugs to measure the appliances. Our results show that by using a richer set of features that are generated by the smart meter itself—and thus no additional computation is needed— we can considerably improve the performance of the methods. The improvement is mainly located at the transient phase where a lot of false negatives are generated by the methods when only active power is used. In fact, our experiments show strong indications that by using four out of the total six features that we are considering, one may get results that are almost as good.

The last point is a good start for designing a light-weight energy disaggregation algorithm. By light-weight, we mean in fact a streaming algorithm that can be executed on the low memory and low processing power CPU on the smart meter. Indeed, the computational capabilities that would be required to solve the problem of energy disaggregation on the cloud in the case of a large scale deployment of such smart meters would be tremendous and certainly not cost effective. This is why most (if not all) of the work must be transferred on the edge, that is, on the smart meter. However, this is equivalent to try and solve the problem of energy disaggregation in a streaming environment where memory is limited and the time to process a measurement is also limited. Thus, looking at the best set of features that the smart meter provides without additional computation for energy disaggregation is indispensable for such an approach. As for the results presented in this paper, we intend to extend our implementation further by collecting larger amount of data for more appliances. We intend to extend our results by including cases where more than one device works at the same time or cases of more general appliance function (multi-state or continuously variable). The conclusions were in general encouraging, as each classifier identified, with a high success rate, whether the corresponding appliance was active or inactive by using only this small set of features generated by the smart meter itself.

**Author Contributions:** Formal analysis, C.K.; Investigation, C.K.; Supervision, S.S., S.K. and K.T.; Writing–original draft, C.K., S.S. and K.T. All authors have read and agreed to the published version of the manuscript.

**Funding:** This research received no external funding.

**Data Availability Statement:** Not applicable.

**Acknowledgments:** This research has been co-financed by the European Regional Development Fund of the European Union and Greek national funds through the Operational Program Competitiveness, Entrepreneurship and Innovation, under the call RESEARCH–CREATE–INNOVATE (project code: T2EΔK-00127).

**Conflicts of Interest:** The authors declare no conflict of interest.

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
