# Peer review of "Evaluation of Features Generated by a High-End Low-Cost Electrical Smart Meter"

_algorithms, doi:10.3390/a14110311_

Round 1

Reviewer 1 Report

The authors have incorporated most of the suggested changes, the paper is in much better shape. However, still more things are needed to be done before the paper is ready for publication:

1. All equations in the paper must be given numbers. Starting from Eqn. 1 and till the end. It is standard practice. It doesn't make any sense that some equations are assigned numbers and some are not. Also, all equations must be refereed from the paper or book from where they have been taken. This is a mandatory thing. If the equation is your own then it doesn't require any reference. 

2. The quality of figure 2 still needs to be improved. It is very difficult to read the both axis and labels. The same goes for Figures 9, 11, 13, 15 and 17. 

3. Table caption is written above the table. (see tables 1-7)

4.  There are still only 15 references in the paper which is a very low number for a journal paper. Add some latest references in the paper e.g. for the equations and in the introduction section. 

Reviewer 2 Report

I'm glad that my reviews are quite consistent. I've read through the revised paper and the responses to my raised issues, and I agree that the authors have addressed all the major issues. Thus, I recommend the acceptance of the paper.

Author Response

We really appreciate your response. Thank you for your time and effort.

This manuscript is a resubmission of an earlier submission. The following is a list of the peer review reports and author responses from that submission.

Round 1

Reviewer 1 Report

The purpose of this study is to present some initial results towards this goal by making heavy use of the characteristics of a particular din-rail meter. The paper presents some interesting ideas. However, it is written in a hurry and has not followed the guidelines recommended by the journal. 

1. The headings are not written properly. Some equations are not numbered and not referenced in the text. The parameters given in the equations are not elaborated.  Figures numbers are not in a sequence. Some tables are also given the captions of the figures.   

2. All the figures have poor visible quality. The text in the figures is not easily readable. The axis in the graphs is not labeled.  

3.  The conclusion is too short and doesn't explain all the things done in the research. 

4. The paper also needs a language review. Many sentences have typos and grammatical mistakes.  

5. The experimental design and the results could have written and explained in a better way.  

Reviewer 2 Report

The paper analyzes the importance of the features in the energy disaggregation problem, where the device status (active or inactive) is identified based on the various electrical signal measurements of that device at that point in time. The authors present an approach where this problem is treated as a classification problem and utilize three classical classifiers to solve it: decision tree, random forest, and multi-layer perceptron. The paper's structure is appropriate and the English language used is on par with the journal's high standards. The paper is quite simple and straightforwards, which is not necessarily the problem. But, the main major issue is that the paper does not contain the actual analysis as the title and the abstract promise it.

The issues are the following.

-The phrase 'multivariate' is commonly mistaken for 'multivariable', as is also the case in this paper. 'Multivariate' denotes the analysis where there are multiple outcome variables, but when there are multiple input variables (features), we call this a multivariable problem/analysis. See [1] for more information.

[1] Ebrahimi Kalan, M., Jebai, R., Zarafshan, E. and Bursac, Z., 2020. Distinction between two statistical terms: multivariable and multivariate logistic regression. Nicotine & Tobacco Research.

-Subsections (such as Related work, Our contributions) are not styled as subsection titles but as normal text. Please use appropriate text styles.

-Is there any particular reason why there was just a train/test (60:40) split of data and not cross-validation? With cross-validation, you would get more results and could conduct statistical analysis to validate the proposed approach even better.

-When you have multiple folds, you get multiple results to conduct statistical analysis if there are statistically significant differences in the classification metrics when different sets of features are used. Otherwise, you can only explain the differences due to chance distribution (splitting into train/test sets).

-The paper's primary focus is the feature evaluation regarding their ability to improve the identification of if the device is ON or OFF (binary classification problem). Yet, there are no real feature-importance results presented. As the decision trees were used, you could plot the tree and look at what features are closer to the root of the tree (as these features have a better ability to split the data into 'clean subsets). With random forest, you could use feature importance calculation where the features are evaluated by how many times they appear in individual trees. With MLP, you could use other permutation-based feature importance calculations or even include transparency methods (SHAP, LIME) to gather knowledge on what features contribute the most to the final decisions.

Reviewer 3 Report

Please see the attached pdf.

Kind regards,

Pascal Schirmer
